# The Stressful Memory Assessment Checklist for the Intensive Care Unit (SMAC-ICU): Development and Testing

**DOI:** 10.3390/healthcare10071321

**Published:** 2022-07-16

**Authors:** Karin Samuelson

**Affiliations:** Department of Health Sciences, Faculty of Medicine, Lund University, P.O. Box 157, SE-22100 Lund, Sweden; karin.samuelson@med.lu.se

**Keywords:** critical care, intensive care unit, stressful memories, COVID-19, assessment, screening, checklist

## Abstract

Stressful or traumatic memories of an intensive care stay may lead to long-term psychological morbidity. Memory assessment is therefore essential to aid in the patients’ recovery process. Acknowledging the large cohort of post ICU patients during the SARS-CoV-2 pandemic, a simple tool for the evaluation of ICU memories is needed. The aim of this study was, therefore, to develop and test the validity and reliability of a short stressful memory assessment checklist, including a distress intensity rating scale, for intensive care survivors. The consecutive sample consisted of 309 patients attending an intensive care follow-up consultation in Sweden. A methodological design was used consisting of four phases. The first three concerned construct and content validity and resulted in a 15-item checklist of potential stressful memories with a Likert-type scale including five response categories for distress intensity rating. To fill out the checklist, a median of 3 (2–3) minutes was needed. A test–retest approach yielded weighted kappa values between 0.419 and 0.821 for 12 of the single items and just below 0.4 for the remaining three. In conclusion, the stressful memory assessment checklist seems to be valid and reliable and can be used as a simple tool to evaluate the impact of stressful ICU memories.

## 1. Introduction

It is well known that critically ill patients’ experiences of being in the intensive care unit (ICU) vary from having none or few blurred memories to recalling almost everything from the ICU stay [1,2]. Neutral, indifferent or positive ICU memories may occur but negative, bothersome and frightening memories such as pain, thirst, tube discomfort and delusional memories are more likely to be remembered [3,4]. The existence of stressful and potentially traumatic memories may contribute to substantial psychological distress and lead to the development of long-term psychological morbidity such as posttraumatic stress disorder (PTSD), anxiety or depression [5,6,7]. Previous studies have shown that the number of stressful events remembered is important [8,9]. However, the magnitude of distress intensity is of the greatest significance, i.e., the imprint of one traumatic experience can be sufficient to cause substantial subsequent distress [10,11]. Nowadays, striving towards less sedation and person-centred care for the critically ill [12], further knowledge and understanding of patients’ experiences are essential in order to improve care. Moreover, the SARS-CoV-2 pandemic resulted in significantly more ICU patients in need of long-term mechanical ventilation resulting in extensive rehabilitation periods [13,14]. The screening and evaluation of patients’ stressful memories may therefore be beneficial in order to improve care and aid in the recovery process.

Several measures exist to assess patients’ memories of the ICU stay. An older questionnaire is the Environmental Stressor Questionnaire (ESQ) including 50 items, which focus only on external stressors [15]. The ICU memory (ICUM) tool was developed 20 years ago and has been widely used [16]. It includes a checklist of factual memories, memories of feelings and delusional memories, and uses dichotomy response alternatives. Another measure, the ICU stressful experiences questionnaire (ICU-SEQ) was targeted at patients who had received prolonged mechanical ventilation only and assessed potential stressful experiences concerning the endotracheal tube and the ICU, with the use of a distress intensity rating [17]. The intensive care experience questionnaire (ICE-q) is a 31-item measure that identifies and assesses four domains of the experience, with the use of two response formats: level of agreement and level of frequency [18]. Several more measures for the assessment of ICU stressors exist [19]. However, these tools mostly originate from the abovementioned older measures, and, moreover, they are extensive and therefore time-consuming.

Despite the wide range of available ICU memory tools, considering the large cohort of post ICU patients that emerged during the SARS-CoV-2 pandemic, a shorter tool including distress scoring would have been useful to evaluate patients’ stressful ICU memories. Since critical ill patients are likely to be exhausted and vulnerable for a significant amount of time in the aftermath of their ICU stay, simpler tools may be preferable. Thus, stressors perceived as very distressing can be identified and information and psychological support can be provided. In Scandinavian ICUs, intensive after-care programs are common [20], and a simple assessment can, hence, serve as a tool in the follow-up consultation at an individual level and might aid in the process of identifying potential intensive care issues suitable for improvement. The aim was therefore to develop and test the validity and reliability of a short stressful memory assessment checklist, including a distress intensity rating scale, for ICU survivors.

## 2. Materials and Methods

### 2.1. Design, Setting and Routines

A methodological study design was used for development and testing of the Stressful Memory Assessment Checklist for the ICU (SMAC-ICU). The study was carried out over three years until the pandemic started in 2020, in one general ICU with seven beds, at a university hospital in the south of Sweden comprising close to 900 adult beds (catchment area population of 1 million citizens). Common diagnostic ICU admission categories included primary medical reason, postoperative complications or major surgery and multiple trauma. About 80% were emergency admissions. The patients were not physically restrained, were never left alone by the staff and visits were allowed round the clock. During invasive mechanical ventilation the sedation goal aimed to keep patients alert and calm or lightly sedated; if not, heavy sedation was required for medical reasons. Patient ICU diaries were usually commenced in the first few days. All discharged patients that had an ICU length of stay of more than 48 h, and their next of kin were enrolled in the nurse-led intensive care after-care programme established in 2005 [21]. A follow-up consultation was held approximately 2–3 months after ICU discharge, and focused on the patients’ and their relatives’ experiences and perceptions of the ICU stay and their current well-being.

### 2.2. Sample and Data Collection

The sample was consecutive and patients eligible were those attending the follow-up consultation and ≥18 years of age. Only Swedish-speaking patients without cognitive impairment were included. The data collection was performed at the end of the consultation by use of the checklist. Patients completed the rating of the stressful memories by themselves, or if they so wished, the items on the checklist and the response categories were read aloud by the after-care nurse. The development and testing of the SMAC-ICU consisted of four phases as described below; the first three concerned validity and the fourth reliability.

In *phase one* the concept and the main domain were outlined, i.e., construct validity [22]. Inspired by the findings of a previous study [3], a preliminary 23-item checklist was formed including the most common distressful memories. Several empty lines were added for patients to add experiences other than those listed. To rate the intensity of distress a Likert scale with 5 response categories was used. At first, the patient was asked: do you recall anything stressful or unpleasant from your intensive care stay? At the end of the checklist, the following question was posed: during the last week, have you been bothered or distressed by your ICU memories? The preliminary checklist was piloted in a sample of 85 patients.

In *phase two* a 14-item checklist including 4 response categories was used. In order to ensure that the revised checklist adequately covered patients’ stressful memories in terms of representativeness, i.e., content validity [23], the checklist was tested in a consecutive larger sample of patients attending the follow-up consultation.

In *phase three*, a 15-item checklist was used including one version with the 4 response categories and one version with a Numeric Rating Scale (NRS, score 0–10). Field-test interviews were carried out in 16 patients in order to capture and evaluate the patients’ judgement of the checklist and the rating scale used, i.e., face validity [22]. After completing the two versions of the checklist, patients were asked about the instructions included, legibility, content and layout of the checklist and the preferred version of response categories.

In *phase four*, the reliability of the 15-item SMAC-ICU with 5 revised response categories was tested with a test–retest approach. During the study period, patients who had completed the checklist (test) were sent a letter two weeks later with a request to fill out the checklist once again (retest).

### 2.3. Data Analyses

After-care statistics and patient characteristics were retrieved from the local ICU register. The checklist results in terms of distress scoring were considered to have ordinal properties, and, hence, were presented in median and 25–75 percentiles (interquartile range), but for overall descriptive purposes, mean and SD were also used. For the test–retest data, change over time in the paired data (sum score) was computed using the Wilcoxon signed-rank test. Agreement between test and retest was calculated using kappa statistics for nominal data (dichotomizing the response categories) and quadratic weighted kappa (Κ*_w_*) with 95% CI for ordinal data [23]. For item level data, Κ*_w_* values ≥ 0.4–0.5 were considered acceptable [24]. Statistical analyses were conducted using PASW Statistics 18 (SPSS Inc., Chicago, IL, USA) and vassarstats.net. Statistical significance was set at *p* < 0.05 (two-tailed).

## 3. Results

### 3.1. Phase One: Test of the 23-Item Checklist

Of the 85 patients included in the test of the preliminary checklist, 79% reported at least one stressful memory of the ICU, and the frequency of recall of the different items listed varied between 2% and 45% (Table 1). About 40% of the patients needed more than 10 min to complete the checklist; a reduction in the number of items and response alternatives was therefore considered important. Revising the checklist, similar items were merged and items remembered by less than 10% and those graded as less than moderately stressful were excluded. Hence, the revised checklist consisted of 14 items. Moreover, the five response alternatives were reduced to four: not at all, a little bit, moderately and a lot.

### 3.2. Phase Two: Test of the 14-Item Checklist

Among the 177 patients, 60% were men, the mean age was 65 ± 14 and the median ICU stay was 4 (2.5–8.5) days. Stressful memories were reported by 122 (70%) patients, with the number of recollections ranging from one to a maximum of 12 memories. All but one item were remembered by 12% or more of the patients (Table 2). The item “*Demeanour of staff unsatisfactory*” was recalled by 5% but was graded as one of the most stressful and was therefore retained. Stressful memories other than those listed, reported in less than 5% of patients and graded as moderately or less stressful, were not added. Fifteen percent of the patients described overwhelming experiences on the body or mind that did not fit into any of the listed items, and a new item “*Overwhelming experience*” was therefore added to the checklist.

### 3.3. Phase Three: Field-Test Interviews of the 15-Item Checklist

Among the 40 patients screened for eligibility, 16 patients with stressful memories of the ICU were included (10 no ICU memories, 6 cognitive impairment, 2 no stressful memories, 2 non-Swedish speaking, 2 communication difficulties, 2 no consent). The items presented were considered relevant by all 16 respondents, and suggestions for additional items were few. Instructions, legibility and the layout of the checklist were perceived as good by all but one. Nine of the 16 respondents preferred the four response category version of the checklist rather than the NRS scale. For both versions, the respondents used a median of 3 (2–3) minutes to fill out the checklists. Some respondents who preferred the NRS version argued that four response categories were too limited, and others concluded that the response alternative “A lot” was not strong enough since some of their memories were experienced as extremely stressful. These respondents also felt it was important to report how their distress was relieved. Thus, phase three resulted in no alterations of the items listed, and the extension to five response categories (*no recall or not stressful, slightly, moderately, quite a bit, extremely),* a revised introductory question (*do you recall anything stressful or unpleasant from your ICU stay?)* and a final question about distress reduction *(do you recall anything that relieved your distress or facilitated your ICU stay?*) (Table A1).

### 3.4. Phase Four: Test–Retest of the 15-Item Checklist

Of the 70 patients eligible, 41 patients completed the first test (10 cognitive impairment, 7 follow-up consultation by telephone, 4 no consent, 3 with PTSD, 3 not Swedish speaking, 2 palliative care). Among the 41 patients, 61% were men, the median age was 65 (47–65) and the median length of ICU stay 4 days (2–6.5). The number of completed retest forms returned was 31 and the median interval between test and retest was 18 (16.4–19) days ranging from 14 to 36. There were no significant differences in characteristics of the non-respondents vs. respondents. Test and retest results showed that no recall occurred in four vs. eight respondents, and the total sum score was a median of 10 (2–16) vs. a median of 8 (0–20), *p* = 0.896. Recall of having highly stressful memories (“quite a bit” or “extremely”) or not, resulted in a kappa value of 0.644 (95% CI; 0.358–0.930) between the two time-points. Among the 15 single items, 12 yielded Κ*_w_* values between 0.419 and 0.821 and three below 0.4 (Table 3).

## 4. Discussion

By definition, critical illness requiring intensive care treatment is a significant stressful event including a wide range of stressors. It is a fact that despite the efforts made towards early comfort using analgesia, minimal sedatives and maximal human care [12], having unpleasant and frightening memories of the intensive care stay is still predominant [13,25]. The SARS-CoV-2 pandemic has shown us that this is especially true for those patients requiring sedation and long-term mechanical ventilation [26,27]. From the patient’s perspective, evaluating the impact of these memories is important, as early stressful and frightening memories might contribute to long-term psychological morbidity after ICU [9]. Moreover, the large cohort of post ICU patients that the COVID-19 pandemic created, and continues to create, requires structured follow-ups and the availability of further rehabilitation interventions [28,29]. To aid in the patient’s recovery process, systematic screening may, thus, be useful as part of a strategy for personalized targeted care and interventions to reduce later psychological distress [9,29]. For example, those memories the patient recalls as highly or extremely stressful need to be explained and discussed with the patient. Providing information and psychological support as well as identifying further needs (and referrals to healthcare professionals if needed) are essential for the patient and their next of kin to manage their distress [21].

The results showed that the checklist is acceptable for use with patients with stressful memories of the ICU, but that it is not perfect in terms of validity and reliability. The first phase of this study clearly showed that the number of items in the checklist had to be reduced. Critically ill patients are often exhausted and may suffer from cognitive impairments and a short and simple checklist was therefore considered paramount. The second phase included a larger sample to test the representativeness of the checklist content, and this revealed that the checklist items were relevant and that most items were remembered as highly stressful. Many of the items presented were also similar to those presented in previous studies concerning stressful memories, showing that memories such as thirst, tube discomfort and delusional memories are commonly remembered as distressing [17,30].

The interviews performed in the third study phase showed that the response alternative “A lot” was not strong enough. Looking at all data, it is evident that ICU survivors are likely to score most of the checklist items as highly stressful, which emphasizes the importance of the distress intensity rating. A dichotomic scale such as Yes or No, or response formats in terms of the level of agreement and level of frequency do not capture the amount of distress experienced. Accordingly, when using the SMAC-ICU, if the patient affirms having been distressed by their ICU memories in the preceding week, it is recommended that the patient is screened for the presence of PTSD-related symptoms with an established screening questionnaire [6,7]. The respondents interviewed also felt that the means of stress relief ought to be reported in the questionnaire. The addition of the last question thereby gives the healthcare staff valuable information about issues important for patient comfort and well-being.

The test–retest results showed that the recall of stressful memories, to some extent, changed over time. For example, fewer patients remembered anything stressful after the two-week period, which is in line with a previous follow-up study using the ICUM tool [11]. Still, the overall stability of the checklist in terms of agreement between the two time-points was satisfactory. The test–retest results of the checklist were acceptable, yielding kappa values about 0.6 and weighed kappa values above the suggested criterion of 0.4 in 12 of the 15 single items.

*Strengths and limitations* High validity and stability of a memory checklist is close to impossible to achieve. The subjective nature of memory experiences makes the accuracy of patients’ recollections difficult to assess; it is a unique experience and therefore a predefined memory checklist is not likely to fit all. Moreover, some memories may have been lost, and others may emerge only later, and information received may have influenced the patients’ recollection at the time of data collection. The fact that the data collection was carried out at the end of the follow-up consultation can be considered as either a strength or a potential weakness of the study. The content of the dialogue may have influenced the patients’ recall and distress rating. On the other hand, the conversation held may have elicited some memories otherwise lost. Validity testing of the checklist was carried out in three phases using quantitative and qualitative measures, which strengthens the study. The final response categories for distress rating were chosen with respect to the respondents’ views and are in accordance with recommendations in the literature [31].

The used retest interval of two weeks is typically recommended for reliability studies [23] and was considered suitable for the ICU respondents. However, the data collection mode of the test and retest differed, which may have influenced the results. In this study a Κ_w_ criterion of >0.4–0.5 was used, which refers to single items [24]. The more often used criterion of >0.7–0.8 refers to summed multi-item scales [23]. A potential weakness in the test–retest is the relatively small sample size of 31. Larger samples are recommended by some, but smaller sample sizes of 15–20 have also been suggested as sufficient [24].

A potential limitation might be that the checklist was not tested in critically ill COVID-19 patients. However, recent research shows that these patients have a similar recall of stressful memories to that of the common ICU population [26,27]. Furthermore, ICU memory assessment is difficult to perform if the patient suffers from cognitive impairment. This suggests that the results of this study, and thereby the SMAC-ICU, may be limited for use in general ICUs only, ICUs with similar intensive care routines as in Scandinavia and in cognitively intact post ICU patients with memories of their ICU stay.

## 5. Conclusions

The SMAC-ICU has satisfactory validity and reliability properties and can be used as a simple tool at an individual level to identify and assess stressful memories of the patients’ ICU stay. The 15-item checklist is short and only takes 2–3 min to fill out, and the open-ended questions at the end make it possible to identify additional distressing memories, and interventions that may relieve distress. Further testing is recommended for use in COVID-19 patients. Thus, the checklist is not a fixed entity and should be tested and, if needed, further revised if changes in the health care setting occur, i.e., in accordance with patients’ responses and the care given. When adapted to the specific context, if used systematically and continuously, it might aid in the process of identifying potential intensive care issues that are suitable for improvement.

## Figures and Tables

**Table 1 healthcare-10-01321-t001:** Phase one. Frequency and rating of stressful memories in intensive care patients by use of the preliminary 23-item checklist with 5 response categories (N = 85).

	*n* (%)	Median (25–75 Percentile) *	Mean Score (SD) *
No ICU recall	18 (21)		
*Stressful memory items*			
Thirst	38 (45)	3 (3–4)	3.2 (±0.78)
Strange experience	28 (33)	4 (1–4)	2.8 (±0.78)
Constraining tubes/devices	25 (29)	3 (1–4)	2.5 (±0.78)
Breathing difficulty	24 (28)	4 (3–4)	3.6 (±0.78)
Communication difficulty	24 (28)	3 (2–4)	2.9 (±0.78)
Full-face mask	24 (28)	3 (2–4)	2.9 (±0.78)
Disturbing interferences	20 (24)	3 (2–4)	2.9 (±0.78)
Sleeping difficulty	18 (21)	2 (2–4)	2.4 (±0.78)
Terrifying dreams	17 (20)	4 (4–4)	3.8 (±0.78)
Panic	16 (19)	4 (3–4)	3.6 (±0.78)
Mouth care	16 (19)	0 (0–2)	0.8 (±0.78)
Pain	16 (19)	3 (2–4)	2.9 (±0.78)
Hallucinations	15 (18)	4 (3–4)	3.3 (±0.78)
Uncertainty/helplessness	14 (16)	3 (3–4)	3.4 (±0.78)
Paranoid delusions	11 (13)	4 (4–4)	3.5 (±0.78)
Turning in bed	11 (13)	0 (0–0)	0.7 (±0.78)
Airway suctioning	10 (12)	2 (1–3)	2.0 (±0.78)
Fear	8 (9)	3 (2–4)	3.0 (±0.78)
Confusion	6 (7)	2 (2–4)	2.0 (±0.78)
Disrespectful staff	4 (5)	2.5 (1–4)	2.5 (±0.78)
Hostile surroundings	3 (4)	0 (0–4)	1.3 (±0.78)
Lack of attention	3 (4)	3 (3–4)	3.3 (±0.78)
Non-caring management	2 (2)	2.5 (2–3)	2.5 (±0.78)
*Other experiences*			
Anxiety	10 (12)		
Having to wear diapers	3 (4)		
Sensation of being trapped	3 (4)		
Nausea	2 (2)		
Disturbed by other patients	2 (2)		
Other	7 (8)		
*Distressed by memories last week*	7 (8)		

Level of distress rating; response alternatives used: 0 = not at all, 1 = a little bit, 2 = moderately, 3 = quite a bit, 4 = a lot. * The median and mean score were calculated among those in recall of the item.

**Table 2 healthcare-10-01321-t002:** Phase two. Frequency and ratings of stressful memories in intensive care patients by use of the 14-item checklist with 4 response categories (N = 177).

	*n* (%)	Median (25–75 Percentile)	Mean (SD) *
No memories of the ICU stay	55 (31)		
*Stressful memory items*			
Terrifying unreal experiences, hallucinations or dreams	71 (40)	3 (2–3)	2.5 (±0.88)
Thirst	66 (37)	3 (2–3)	2.5 (±0.79)
Communication difficulty	51 (29)	3 (2–3)	2.5 (±0.76)
Breathing difficulty	49 (28)	3 (3–3)	2.8 (±0.52)
Tube discomfort	47 (27)	3 (2–3)	2.2 (±0.97)
Full-face mask discomfort	39 (22)	3 (2–3)	2.4 (±0.85)
Pain	39 (22)	3 (2–3)	2.4 (±0.68)
Anxiety/Fear	35 (20)	3 (3–3)	2.7(±0.51)
Panic	34 (19)	3 (3–3)	2.9 (±0.36)
Disturbing noise or conversation	27 (15)	3 (1–3)	2.3 (±0.94)
Unpleasant procedures	26 (15)	2 (0–3)	1.5 (±1.27)
Difficulty sleeping or resting	25 (14)	3 (2–3)	2.4 (±0.71)
Airway suctioning	21 (12)	2 (1–3)	2.0 (±1.00)
Demeanour of staff unsatisfactory	8 (5)	3 (3–3)	2.9 (±0.35)

Level of distress rating; response alternatives used: 0 = not at all, a little bit = 1, moderately = 2, a lot = 3. * The median and the mean score were calculated among those in recall of the item.

**Table 3 healthcare-10-01321-t003:** Test and retest scores of stressful memory items and corresponding weighted kappa values (*K_w_*) in intensive care survivors (N = 31).

Stressful Memory Items	Test, Sum Score	Retest, Sum Score	Test, Median Score	Retest, Median Score	*K_w_*	95% CI (*K_w_*)
Thirst	54	50	2 (0–3)	1 (0–3)	0.815	0.711–0.918
Terrifying unreal experiences, hallucinations or dreams	46	41	1 (0–3)	0 (0–3)	0.625	0.361–0.899
Anxiety/fear	35	34	0 (0–3)	1 (0–2)	0.821	0.772–0.923
Communication difficulty	32	30	0 (0–2)	0 (0–2)	0.807	0.768–0.846
Breathing difficulty	31	17	0 (0–2)	0 (0–1)	0.419	0.075–0.762
Panic	24	25	0 (0–1)	0 (0–2)	0.693	0.400–0.988
Pain	22	26	0 (0–1)	0 (0–2)	0.679	0.384–0.974
Tube discomfort	19	21	0 (0–0)	0 (0–1)	0.687	0.349–1.00
Full-face mask discomfort	17	21	0 (0–1)	0 (0–1)	0.380	0.076–0.684
Disturbing noise or conversation	15	14	0 (0–1)	0 (0–1)	0.735	0.475–0.944
Overwhelming experience	14	21	0 (0–0)	0 (0–2)	0.346	0.023–0.675
Unpleasant procedures	13	16	0 (0–0)	0 (0–1)	0.544	0.085–1.00
Difficulty sleeping or resting	11	11	0 (0–0)	0 (0–1)	0.379	0.054–0.703
Airway suctioning	10	12	0 (0–0)	0 (0–0)	0.650	0.414–0.885
Demeanour of staff unsatisfactory	1	3	0 (0–0)	0 (0–0)	0.475	0–0.972
Total sum score (all 15 items)	344	342	10 (2–16)	8 (0–20)		

Level of distress rating; response alternatives used: No recall or not stressful = 0, a little bit = 1, moderately = 2, quite a bit = 3, extremely = 4. The analyses are calculated for all 31 participants, including those without recall of any stressful memory.

## Data Availability

The datasets generated and analysed in this study are not publicly available due to their containing information that could compromise the privacy of the research participants but are available from the corresponding author upon reasonable request.

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
