# Peer review of "The Stressful Memory Assessment Checklist for the Intensive Care Unit (SMAC-ICU): Development and Testing"

_healthcare, 2022, doi:10.3390/healthcare10071321_

Round 1

Reviewer 1 Report

Thank you for giving me this opportunity to review this interesting paper. The paper concerning The Stressful Memory Assessment Checklist for the Intensive Care Unit (SMAC-ICU); Development and Testing. The paper was well written and study designs and methodology was well constructed and performed. Only  couple things need to be addressed:

-           The relational behind developing a new assessment tool still unclear to me. The provided rational by others is not convincing and dose not have a clear clinical contribution. Therefore, authors need to elaborate in this perspective in the introduction and discussion sections.

-           Tables are overlapped in the versions I have seen; therefore, authors need to be reformatted properly.  

-           In the discussions, limitations were not included and discussed.

Author Response

-        The relational behind developing a new assessment tool still unclear to me. The provided rational by others is not convincing and dose not have a clear clinical contribution. Therefore, authors need to elaborate in this perspective in the introduction and discussion sections.

Answer; Thank you very much for your comments. I agree to some extent. In the manuscript I have outlined that the existing tools are extensive and old and therefore a newer and shorter tool is needed. I have now tried to clarify it further in the introduction and discussion and conclusion section. Maybe it is evident to me, as I have been working in a ICU follow-up service for a long time, but for the reader I agree that the benefits of using an assessment tool needed to be explained better.  

-           Tables are overlapped in the versions I have seen; therefore, authors need to be reformatted properly.

Answer; Yes, there is a problem with the Tables that move around. I have asked the editorial office for help with this. If I try to move them now it causes further chaos. 

-           In the discussions, limitations were not included and discussed.

Answer; I think I do have discussed several limitations of the study but I have now inserted a headline “Strengths and limitation” so it becomes more clear to the reader.

Reviewer 2 Report

The article hereby analyzed postulates a novel questionnaire model for patients recently dismissed from an ICU unit with the goal of consolidating the character and nature of specific stressors coming from this surrounding environment, besides the level of distress provoked by each specific factor. The accountability for potential long-term consequences such as PTSD (post-traumatic stress disorder) harbored by highly stressful, isolated events most likely represents the actual novel feature that the authors consider and further introduce to the readers, as outlined by a short analysis of similar studies (ESQ, ICUM, ICU-SEQ, ICE-q) of the literature performed in the introduction paragraph - previous studies tend to neglect the magnitude of distress for single events, instead enhancing the role of highly prevalent stressors regardless of their gravity.

Data gathering and preparation is transparently illustrated in order to lower bias of interpretation from the very first phases of the study, at expense of the total cohort size that turns out to lower progressively as the interviews advance from the first to the fourth phase. Care has been taken also for communication barriers, for example by deciding to exclude patients with insufficient cognitive function and individuals not speaking local language (in this case, Swedish). Chances for a consistent usage for the tool currently prototyped were also a concern, this being the reason behind the continuous skimming of unlikely relevant complaints and blending of similar ones in between themselves with the goal of significantly lower the average time for questionnaire’s completion.  Worth of note is the fact that the questionnaire template and structure evolved consistently also dependently from the ICU survivors’ feedback, raising the consistency and validity of the model – this is demonstrated by the last line ultimately introduced by the authors at the end of the phase four’s table, regarding potential suggestions that former critically ill patients had for means of help and relief that could target individuals in the same condition. In this same phase a test-retest approach was used to address variability of response for the same memory in a given individual, in an attempt to address and postulate individually-tailored post-ICU rehabilitation’s program.

After-care statistics and patient characteristics were retrieved from the local ICU register. The checklist results in terms of distress scoring were presented in median and 25-75 percentiles, mean and SD were also used. For the test-retest data, change over time in the paired data (sum score) was computed using the Wilcoxon signed-rank test. Agreement between test and retest was calculated using kappa statistics for nominal data (dichotomizing the response categories) and quadratic weighted kappa (Κw) with 95% CI for ordinal dataFor item level data, Κvalues ≥0.4–0.5 have been considered acceptable . Statistical analyses were conducted using PASW Statistics 18 (SPSS Inc., Chicago, IL, USA) and vassarstats.net. Statistical significance was set at p<0.05 (two-tailed).

Besides the aforementioned illustrated benefits and perks of this questionnaire model, awareness should be raised regarding its implications especially after the COVID-19 global pandemic, which led to a tremendous increase in the number of ICU survivors that had no cognitive impairment, and therefore could more clearly recall the traumatic experiences that they subjectively found as stressful during their hospital stay. Moreover, as outlined by the authors, countries as Sweden offer specific programs of rehabilitation for such patients that, with the help of this individual questionnaires, can be better tailored according to each former patient,leading inevitably to positive adjustments in cost-effectiveness ratios. This is clearly advantageous for institutions ready to make an investment in order to obtain long term benefits in terms of hospital stay-related morbidity reduction. Far from being perfect, these models need further research, bigger cohorts and more awareness raised on them, but this questionnaire template hereby proposed represents a good example and additional effort should be implemented in order to address its weaknesses.

Author Response

Answer; I thank you for your review and the excellent summery of my manuscript. I agree with your reflections about the importance of obtaining long term benefits (morbidity reduction) for our patients– and models such as questionnaires or checklists could play an important role to capture the individual response as well as the overall picture – but as you say, we need to test these models in bigger cohorts. Concerning ICU patients, follow-up services are a good place to start using these models.

Reviewer 3 Report

Dear Author,

Thank you for your manuscript. The manuscript entitled “ The Stressful Memory Assessment Checklist for the Intensive Care Unit (SMAC-ICU); Development and Testing” deals with an interesting and important issue that perfectly fits the journal’s scope. I have only a few minor comments.

I would suggest specifying your study aim: „Develop and test the validity and reliability of the stressful memory assessment checklist“.

Next, in the Methods section, the ethical principles of the study and obtaining the informed consent procedures are not described. Please provide this important information.

Author Response

I would suggest specifying your study aim: „Develop and test the validity and reliability of the stressful memory assessment checklist“.

Answer; This is a good suggestion and the aim has been changed accordingly; The aim was therefore to develop and test the validity and reliability of a stressful memory assessment checklist, including a distress intensity rating scale, for ICU survivors.

Next, in the Methods section, the ethical principles of the study and obtaining the informed consent procedures are not described. Please provide this important information.

Answer; Yes, this is very important information and it is presented in the end of the manuscript according to the guidelines of the journal.

Institutional Review Board Statement: This study was regarded as a quality assurance measure within the department and approval was granted by the head of the department, the ICU consultant in charge and the ICU head nurse. The study was conducted in accordance with the Declaration of Helsinki [25].  

Informed Consent Statement: Before the data collection was started at the end of the follow-up consultation, the respondents were informed about the purpose of the questions, the voluntary aspect of participating, and guaranteed anonymity. Written consent was obtained for all participants except for those suffering from critical illness neuropathi or myopathy; oral consent was then considered sufficient. Data concerning after-care patients were kept confidential, and all the data presented in this study were rendered anonymous by the use of coding.

Reviewer 4 Report

The authors have made an interesting attempt on “The Stressful Memory Assessment Checklist for the Intensive Care Unit (SMAC-ICU); Development and Testing.” The manuscript is interesting; however, the authors need to justify the scientific writing manuscript. Some of the general comments are provided below:

1.     Did the authors conduct a pilot study to validate the clarity and comprehensibility of survey questions, if yes, what is the test score of Cronbach's alpha.

2.     What is the representative minimum sample size required for statistical analysis?

3.     How to take effective measures to respond spontaneously and rapidly to unexpected situations as the COVID-19 Pandemic? More policy implications can be discussed in the conclusion part.

4.     More detailed information about the sample, such as the scale, organization structure, medical level and geographical distribution, should be described in the Data part.

5.     The sample size is very small, it would also be interesting to do such a study on a larger scale with patients from different classes of society based on their social status.

6.     Are the conclusions applicable to other countries? The significance of the paper needs to be elaborated. 

Author Response

  1. Did the authors conduct a pilot study to validate the clarity and comprehensibility of survey questions, if yes, what is the test score of Cronbach's alpha.

Answer; Thank you for your comments. We did not conduct a pilot study because we previously conducted a larger study including 250 patients, asking the patients about what they experienced as unpleasant or distressing ( see reference nr 3). So we used the results of that study to form the first draft of the SMAC-ICU. We have described this in the Sample and datacollection, phase one;  Inspired by the findings of a previous study [3], a preliminary 23-item checklist was formed including the most common distressful memories.

Cronbach´s alpha concerns internal consistency of an instrument that measures a phenomenon, trait, behavior or symptom; for example Health-related quality of life, or pain or cognitive impairment. Then all questions in the instrument are related/correlated to some degree. Thus, if the Cronbach´s alpha value is high, all questions mirrors /captures the phenomena well. Our stressful memory assessment checklist (SMAC-ICU) is not an instrument measuring a phenomenon. It is a checklist that covers different possible distressing memories or events. It covers several phenomena such as pain, anxiety, demeanour of staff and so on. Therefore we did not test for Cronbach´s alpha.

  1. What is the representative minimum sample size required for statistical analysis?

Answer; this is an interesting question but very hard to answer. The sample size required depends on what statistic test you want to perform and it also depends on the effect size/power and the significance level you choose to use. In phase one and two we only used descriptive statistics, including 85 and then 177 patients, which we think is a relevant amount when developing a checklist, and in phase 3 we used a more qualitative approach which justifyies including only 16 patients. In phase 4 we used “kappa” statistics and I agree that the sample size here is sparse and this has been outlined in the discussion section; A potential weakness in the test-retest is the relatively small sample size of 31. Larger samples are recommended by some, but smaller sample sizes of 15-20 have also been suggested as sufficient [24].

  1. How to take effective measures to respond spontaneously and rapidly to unexpected situations as the COVID-19 Pandemic? More policy implications can be discussed in the conclusion part.

Answer; Yes, I agree. I have added some sentences in the conclusion part that will clarify the use of the SMAC-ICU in unexpected situations.

  1. More detailed information about the sample, such as the scale, organization structure, medical level and geographical distribution, should be described in the Data part.

Answer; Yes, I have clarified some of your requests in the beginning of the Material and method section. However, further information of the participants’ characteristics then those presented further on in the manuscript, is not available. I only used data accessible on the follow-up consultations, avoiding screening of the patients records due to ethical reasons.

  1. The sample size is very small, it would also be interesting to do such a study on a larger scale with patients from different classes of society based on their social status. 

Answer; Yes I agree. The relationship between how patients experience or recall a hospital stay or an ICU stay may differ based on their social status. Unfortunately, we did not include such data. When the check-list is tested in a larger cohort social data should be included.

  1. Are the conclusions applicable to other countries? The significance of the paper needs to be elaborated. 

Answer; Yes, I have stated this in the Limitation and strengths section; This suggests that the results of this study, and thereby the SMAC-ICU may be limited for use in general ICUs only,  with similar intensive care routines as in Scandinavia, in cognitively intact post ICU patients with memories of their ICU stay.

Do you need further elaboration?

Round 2

Reviewer 4 Report

The authors' response is satisfactory so the article can be published now.